# Increase in Virus-Specific Mucosal Antibodies in the Upper Respiratory Tract Following Intramuscular Vaccination of Previously Exposed Horses Against Equine Herpesvirus Type-1/4

**DOI:** 10.3390/vaccines13030290

**Published:** 2025-03-10

**Authors:** Bettina Wagner, Christiane L. Schnabel, Alicia Rollins

**Affiliations:** Department of Population Medicine and Diagnostic Sciences, College of Veterinary Medicine, Cornell University, Ithaca, NY 14853, USA

**Keywords:** equine herpesvirus, mucosal immunity, antibody, frequent vaccination, upper respiratory tract, inactivated vaccine, IgG4/7, IgG1, IgA

## Abstract

Background/Objectives: Equine herpesvirus type-1 (EHV-1) enters through the upper respiratory tract (URT) and causes respiratory disease, abortions, and myeloencephalopathy in equids. Pre-existing immunity at the viral entry site, especially mucosal IgG4/7 antibodies, has recently been shown to correlate with protection from disease and incomplete viral replication at the URT. Here, we tested whether intramuscular (i.m.) vaccination with a commercial inactivated EHV-1/4 vaccine can induce mucosal antibodies (mucAbs) at the URT. Methods: Adult horses with complete EHV-1 vaccination and/or exposure histories were vaccinated i.m. six times within eight months. Before and after each vaccination, blood and nasal swab samples were obtained. Serum and mucAbs were measured in fluorescent bead-based EHV-1 assays. Results: All horses still had existing EHV-1 specific serum and mucAbs prior to vaccination, which were mainly composed of IgG4/7 antibody isotypes. Serum IgG4/7 significantly increased after the first vaccination and stayed high until the end of the study. An additional short-lasting serum IgG1 response was only induced by the first vaccine application. At the URT, mucAbs increased after five out of six i.m. vaccine injections. Like the systemic antibody response, mucAbs were dominated by IgG4/7 and a small IgG1 increase after the first vaccination. Conclusions: Our data emphasize that robust EHV-1 specific mucAb levels are obtained after i.m. vaccination with the inactivated EHV-1/4 vaccine used here. The findings have important implications for evaluating EHV-1/4 vaccines for their ability to induce and maintain protective mucosal IgG4/7 antibodies.

## 1. Introduction

Equine herpesvirus type-1 (EHV-1) is an alphaherpesvirus and is highly prevalent in most equine populations worldwide [1,2,3,4,5]. EHV-1 is a highly host-specific respiratory virus that causes mild to moderate respiratory disease in most equids [6,7,8]. However, EHV-1 infection can also cause neonatal death, severe respiratory infection in young horses, abortion in late pregnancy [5], and/or equine herpesvirus myeloencephalopathy (EHM) in adult horses [9,10,11]. Due to medical costs, loss of foals and affected horses, quarantine, and interruptions in training, horse travel, and equine events, the equine industry is majorly impacted, both economically and logistically, by abortions and EHM outbreaks [2,12]. Like other herpesviruses, EHV-1 establishes latency after infection and all clinical manifestations can occur during recrudescence [1,2,13]. Commonly, foals or weanlings first become infected early in life by their dams or herd mates. In young horses, disease can be unrecognized or mild. However, these now latently infected horses become reservoirs of the virus and can reactivate EHV-1 in periods of stress [1,2,13,14]. EHV-1 transmission requires close contact. It can occur directly via respiratory secretions by nose-to-nose contact [9] or, indirectly, via contact with nasal secretion-contaminated fomites. The viral entry site of EHV-1 is the upper respiratory tract (URT). The virus first enters respiratory epithelial cells and replicates in the nasal mucosa [7,8,15,16]. It then enters leukocytes in the nasal and retropharyngeal lymphoid tissues to next spread systemically via cell-associated viremia [17]. Viremia typically occurs between days 4 to 13 after infection [15,16]. It is currently assumed that EHV-1 latency is established during viremia in the trigeminal ganglia, in ganglionic neurons, and/or in leukocytes [18,19,20,21]. The viremic phase is also the accepted prerequisite for enabling the spread of the virus from infected leukocytes to vascular endothelial cells in the pregnant uterus and/or the central nervous system [2,22]. The viral spread to vascular endothelial cells during EHV-1 infection is the key event leading to neonatal foal death, abortion in late pregnancy, or EHM in adult horses [10,11,22]. Abortions caused by EHV-1 are estimated to happen in >50% of infected pregnant mares, while clinical signs of EHM are reported in up to 10% of infected adult horses during outbreaks [2].

In the United States (US) and Europe, several EHV-1/4 vaccines are available and labeled to reduce respiratory disease and/or abortions caused by EHV-1 and EHV-4. Vaccines include inactivated and modified-live virus vaccines [23,24]. The use of EHV-1/4 vaccines and improved management practices have reduced the occurrence of abortion storms. However, despite the availability of EHV-1/4 vaccines, EHV-1 outbreaks have continued to occur [2,8,11]. In the US, EHV vaccination is currently not included in the core vaccine recommendations for horses [25] and is therefore not performed on a regular basis in most of the horses. Categorized as a risk-based vaccine, EHV-1/4 vaccination of the US horse population varies from pregnant mares being vaccinated 3–4 times during gestation, to twice annually (every 6 months) for sports horses, to infrequent vaccination or not vaccinated at all. The first EHV-1/4 vaccination is generally recommended between 4 and 6 months of age. The first dose should be followed by two booster vaccinations with 4–6 weeks between the first and second doses and the third dose at 10–12 months to establish initial immunity [26]. The vaccination recommendations are identical for inactivated EHV-1/4 and modified-live EHV-1 vaccines in adult non-pregnant horses, while broodmares are often vaccinated more frequently [27]. Overall, EHV-1/4 vaccination practices vary widely depending on the risk of infection and/or requirements of regional or national equine organizations [27].

Protective immunity against EHV-1 is complex and composed of local immunity at the URT [8,28,29], systemic antibody [28,29], and peripheral T-cell responses [30]. Systemic EHV-1 antibodies, in particular IgG4/7, correlated with protection from disease, nasal shedding, and cell-associated viremia [28,29]. Peripheral EHV-1 specific cytotoxic T-cell responses were associated with protection from EHM, the neurologic form of the disease [30], and abortions [31]. Peripheral EHV-1 specific interferon-gamma (IFN-γ) producing CD8^+^ T-cells and CD4^+^ T-helper 1 (Th1) cells were induced after EHV-1 infection and vaccination [32,33,34,35], and their proportions in peripheral blood vary with time after infection, age, pregnancy, and vaccination [34,35].

More recently, the crucial role of intranasal mucosal immunity as a first line of defense against EHV-1 at the URT entry site of the virus has been identified. Both EHV-1 specific systemic and mucosal antibodies (mucAbs) correlated with protection from infection, disease, nasal shedding, and cell-associated viremia in horses [28,29]. In more detail, protective immunity at the URT is characterized by pre-existing EHV-1 specific IgG4/7 mucAbs and their rapid increase after EHV-1 infection [28,29]. These EHV-1 specific IgG4/7 mucAbs are highly effective in neutralizing EHV-1 at the viral entry site and substantially lower the viral load at the URT [36]. After experimental EHV-1 challenge of immune horses with existing mucosal IgG4/7 antibodies, EHV-1 replication in the respiratory epithelium is incomplete and progression to the viremic stage is largely inhibited [36]. In addition, immune and protected horses lack secretion of mucosal antiviral IFN-α and inflammatory markers, including the chemokines CCL2 and CCL3, and soluble CD14 (sCD14) [28,29], all of which are secreted in high concentrations in non-immune horses that are susceptible to EHV-1 infection and develop fever, clinical signs, nasal shedding, and cell-associated viremia [15,16,28]. The opposite IFN-α secretion pattern in immune and non-immune horses is further supported by high mucosal upregulation of the genes encoding IFN-inducing proteins IFIT2 and IFIT3 in non-immune horses, which are not upregulated after EHV-1 infection of immune horses [25]. However, immune horses rapidly upregulate gene expression and mucosal secretion of the homeostasis regulator antileukoproteinase, also known as secretory leukocyte protease inhibitor (SLPI) [37] and secrete mucosal anti-inflammatory IL-10 and the T-cell chemokine CCL5 after EHV-1 challenge [28,29]. Overall, this emphasizes that immune horses can rapidly interfere with EHV-1 infection, neutralize and destroy the virus at the URT, and orchestrate the mucosal antiviral defense by balancing inflammation, preventing antiviral danger signals, and regulating mucosal homeostasis despite EHV-1 exposure. These new findings support that mucosal immunity is highly effective in preventing disease after EHV-1 infection and, in fully protected horses, also inhibits viral shedding and cell-associated viremia, the prerequisite for EHM [28,29,36,37]. Thereby, the accumulated evidence suggests that horses with strong mucosal immunity are unlikely to establish EHM. This has implications for EHV-1 vaccine design, vaccine efficacy testing, and for evaluating existing EHV-1 and EHV-1/4 vaccines for their potential to induce mucosal EHV-1 specific IgG4/7 antibodies at the URT.

Here, we performed a vaccination study using a commercial inactivated EHV-1/4 vaccine labeled for intramuscular (i.m.) injection. The same inactivated EHV-1/4 vaccine was previously used in EHV-1 naïve, pregnant mares and resulted in short-lasting and sometimes declining systemic antibody values after frequent vaccination [35]. During this study, a similar vaccination frequency was used in adult, non-pregnant horses with pre-existing EHV-1 immunity (pre-immune). The two goals of this study were to explore (1) whether the i.m. vaccine application route induces mucosal antibodies against EHV-1, and (2) if non-pregnant, adult horses (mares and geldings) also produce short-lasting systemic antibody responses after frequent vaccination.

## 2. Materials and Methods

### 2.1. Horses

Fourteen adult horses of the Icelandic research horse herd at Cornell University were enrolled in this vaccination study. All the horses had a complete EHV-1/4 vaccination and/or EHV-1 infection history since they were born. These adult research horses had similar EHV-1 exposure histories to many adult horses presented to veterinarians in the US. Six of the horses were previously EHV-1/4-vaccinated (Prev-Vacc group). The other eight horses were previously experimentally infected with EHV-1 (Prev-Infect group). All the horses in the Prev-Vacc group were mares, 9–13 years of age, and their last EHV-1/4 vaccination was performed 22 months prior to this frequent vaccination study (Table 1). The Prev-Infect group was composed of five geldings and three mares, all 5 years of age, and their last prior EHV-1 infection happened 21 months before this vaccination approach (Table 1).

All the horses lived outside in herds of 6–12 horses. They were kept on pastures with run-in sheds, grazing in the summer and fed grass hay in the winter. Horses had free access to water and mineral salt blocks. For sampling and vaccination, the horses were brought into a barn with individual box stalls for a short period of time. They were restrained on a halter and lead rope for the procedures. Sedation was not needed for these trained research horses. After each sampling and vaccination, the horses were again turned out to their pastures with their herd mates. All the horses survived the study.

### 2.2. EHV-1/4 Vaccination

Horses were vaccinated with an approved, inactivated EHV-1/4 vaccine (Calvenza EHV^®^, Equine Rhinopneumonitis Vaccine, Boehringer Ingelheim, Vetmedica, St. Joseph, MO, USA). This commercial vaccine was labeled for the vaccination of healthy, susceptible horses 6 months of age or older, including pregnant mares, to aid in the reduction of respiratory disease due to EHV-1 and/or EHV-4. According to the vaccine supplier, the vaccine contains a unique respiratory origin EHV-1 isolate in a Carbimmue^®^ adjuvant system that allows for dual-phase antigen presentation. The same vaccine was used in a previous EHV-1/4 vaccination study in pregnant mares using similar vaccination intervals [35]. Here, the initial two vaccinations were given 21 days apart, on day 0 (d0, V1) and d21 (V2). This was followed by four additional vaccinations at 2 months (2m, V3), 3m (V4), 6m (V5), and 8m (V6), as outlined in Figure 1. All the horses received the same vaccine batch (serial number 3220022A, expiration date 29 May 2017). The vaccination was performed between February and September 2016. The vaccine was given i.m. into the pectoral muscles using 1 inch injection needles.

### 2.3. Sample Collection

Blood and nasal swab samples were collected on days 0, 12, and 21, and then monthly at 2–10 months after the initial vaccination (Figure 1). Blood samples were obtained from the *V. jugularis* and collected into vacutainer tubes without anticoagulant. Tubes were kept at 4 °C overnight and then serum was collected after centrifugation at 500× *g* for 15 min, aliquoted and stored at −80 °C until serum antibody measurements were performed. Nasal swab samples were collected using two polyester-tipped applicators (Puritan Medical Products Company, Gullford, ME, USA), which were swabbed on the nasal mucosa of the left nostril for approximately 5 s. Swabs were immediately put in a polystyrene tube with 1 mL of sterile phosphate buffered saline (PBS). The swabs in PBS were maintained at 4 °C and processed within 2 h after sampling. In the laboratory, nasal swab samples were spun at 1000× *g* at 4 °C for 5 min leaving the swabs in the collection tube. The diluted nasal secretion fluid surrounding the swabs was collected into a 1.5 mL centrifuge tube and was stored at −20 °C until mucAb analysis. On vaccination days, all the samples were always collected prior to vaccination.

In addition, banked serum samples from the time interval between the last EHV-1/4 exposure of these horse (Table 1) and V1 of this vaccination approach were measured for EHV-1 specific antibodies. Samples for this analysis were taken 20 or 21 months, and 15 or 14 months before V1 of this approach from the previous vaccinated or infected horses, respectively. Additional samples from all horses were taken 11, 4, and 2 months before V1.

### 2.4. EHV-1 Specific Antibody Quantification (EHV-1 Risk Evaluation Assay)

EHV-1 specific antibody quantification was performed using the EHV-1 Risk Evaluation Assay (Animal Health Diagnostic Center, Cornell University, Ithaca, NY, USA). This accredited fluorescent bead-based assay measures EHV-1 glycoprotein C (gC) specific antibodies (total Ig) and IgG4/7 isotypes. The original assay was initially validated against EHV-1 serum neutralization tests [35]. An additional assay optimization step was performed to increase the linear detection range of the diagnostic EHV-1 Risk Evaluation Assay [15]. In brief, a monoclonal antibody against equine IL-4 (clone 25) was coupled to fluorescent beads numbered 35 (Luminex Corp., Austin, TX) followed by incubation with recombinant IL-4/EHV-1 gC. IL-4/gC was expressed in mammalian cells [35]. The EHV-1 gC beads were then incubated with serum diluted at 1:400 or with undiluted nasal secretion samples for measuring serum or mucAbs, respectively. For total anti-gC antibody measurements (total Ig), a biotinylated polyclonal goat-anti-horse IgG(H + L) antibody (Jackson Immunoresearch Laboratories, West Grove, PA, USA) was used for detection followed by another detection step with streptavidin-phycoerythrin. The assay was measured in a Luminex 200 analyzer (Luminex Corp., Austin, TX, USA) and quantitative antibody values were reported as median fluorescence intensity (MFI).

In addition to the routine EHV-1 Risk Evaluation Assay measuring anti-gC total Ig, the assay was multiplexed to include anti-EHV-1 gB and gD total Ig quantification in all samples. For multiplexing, fluorescent beads numbered 33 and 36 (Luminex Corp., Austin, TX) were also coupled with the equine IL-4 mAb. Bead 33 was then incubated with IL-4/EHV-1 gB and bead 36 with IL-4/EHV-1 gD [35]. For EHV-1 gC-specific isotype detection in serum and nasal secretion samples, IgG and IgA isotype detection reagents included biotinylated monoclonal antibodies specific for IgG1 (CVS45) and IgG4/7 (CVS39) (both [38]), IgG3/5 (clone 586 [39]), and IgA BVS2 [40,41]. All other steps of the assay remained the same as described above.

### 2.5. Statistical Analysis

Shapiro–Wilk normality tests were performed for serum and mucAb data and showed that most of the antibody data were not normally distributed. Thus, non-parametric analyses were performed. Serum and mucAb responses of the Prev-Vacc and Prev-Infect groups were compared for each time point by repeated measures ANOVA with Šídák’s multiple comparison tests. The latter approach was also used to compare differences between the antibody responses of mares and geldings. Serum and mucAbs in response to vaccination were compared for all horses (n = 14) between the pre-V1 vaccination sampling timepoint, d0, and all later timepoints by repeated measures ANOVA followed by Dunnett’s multiple comparisons tests. The statistical analysis was performed in GraphPad Prism, version 9, and *p*-values of <0.05 were considered significant.

## 3. Results and Discussion

### 3.1. Comparison of Antibody Responses Before and After Vaccination Between Groups and Gender

Systemic (serum) and mucAbs were compared before and after frequent vaccination with an inactivated EHV-1/4 vaccine between the Prev-Vacc and Prev-Infect groups, and between mares and geldings. Differences in the responses between the two groups or between mares and geldings were not observed. Therefore, the antibody responses before and after vaccination were analyzed for all 14 horses together as one group.

### 3.2. Decline in Antibody Responses in the Absence of EHV-1/4 Exposure

Before the start of the frequent EHV-1/4 vaccination, horses were not exposed to EHV-1 or EHV-4 for the past 21 or 22 months (Table 1). During this 22-month period, serum antibodies were measured in the EHV-1 Risk Evaluation assay to evaluate their decline over time (Figure 2). One month after the horses’ last EHV-1/4 exposure (−21/−20 months pre-V1), anti-EHV-1 gC total Ig and IgG4/7 antibodies were high with a median of 12,965 MFI for total Ig (range: 8822–16,543 MFI), and 4322 MFI for IgG4/7 (range: 1558–10,735 MFI). Antibodies declined quickly within the next 6 months and then stayed at an almost stable level for the next 14 to 15 months. At sampling time 0, directly before V1 of the frequent EHV-1/4 vaccination, anti-gC total Ig had declined to a median of 7353 MFI (range: 3357–12,455 MFI) and 882 MFI (range: 124–4080 MFI). Before V1, all horses were still above the EHV-1 Risk Evaluation assay’s total Ig protective cut-off level of 3000 MFI, while 3/14 horses had IgG4/7 values below the 400 MFI cut-off for this isotype.

Overall, these results showed that adult, non-pregnant horses with several previous EHV-1/4 exposures can maintain EHV-1 antibodies above the protective cut-off levels for almost 2 years or longer, depending on the individual response and antibody levels after the last exposure.

### 3.3. EHV-1 Specific Serum Antibody Responses After Frequent Vaccination

We have previously shown that pregnant mares responded to frequent EHV-1/4 vaccination with declining serum antibody levels after several administrations of the vaccine [35]. Here, our goal was to identify how non-pregnant, adult horses (mares and geldings) respond to frequent EHV-1/4 vaccination using the same vaccine and similar vaccination intervals (Figure 1). It should be emphasized that our goal was to provide information about the effects of repeated vaccination on antibody responses. The study was not designed to recommend frequent vaccination practices for horses in the field.

All horses had pre-existing EHV-1 specific serum antibodies prior to the frequent vaccination study performed here (Figure 3A). These pre-existing antibodies were residual antibodies from prior EHV-1/4 vaccination or experimental EHV-1 infection that last happened 22 or 21 months before this study, respectively (Figure 2). Here, serum antibodies against EHV-1 gC significantly increased (*p* < 0.0001) after the first vaccination V1 with the inactivated vaccine by 12 days post-vaccination. Serum antibodies then stayed high for the remainder of the study without major changes after the additional vaccine administrations V2–V6 (Figure 3A). Similar steady serum antibody responses were observed for EHV-1 gB and gD-specific antibodies after an increase from pre-existing antibodies in response to V1 (*p* < 0.0001; Appendix A).

As mentioned above, a similar frequent EHV-1/4 vaccination approach, with the same inactivated vaccine in pregnant mares, resulted in a significant decline in EHV-1 specific serum antibodies after vaccination at 2 and 8 months [35]. This approach showed that the decline in serum antibodies was not reproducible in non-pregnant mares and geldings. Besides pregnancy, another difference between the pregnant mare and this study was the EHV-1 immune status of the horses prior to vaccination. Here, we vaccinated horses with pre-existing immunity, while the pregnant mares in the earlier study [35] were EHV-1 naïve prior to vaccination. The results suggested that declining and/or short-lived antibody responses after frequent EHV-1/4 vaccination might be related to pregnancy and/or are characteristic for horses that have not been exposed to EHV-1 antigens previously. Last, it should also be noted that two different vaccine batches were used for the pregnant mare vaccination and this study, which could have contributed to the different outcomes between the studies, although this seems less likely than the reasons stated above. Nevertheless, our findings overall suggested that the robustness and longevity of an EHV-1 antibody response is likely influenced by previous exposure to EHV-1 antigens. The rapid increase in serum IgG4/7 antibodies in our horses > 20 months after their last EHV-1 infection or vaccination supported the existence of a long-lived, EHV-1 specific IgG4/7 memory B-cell population. Therefore, frequent vaccination seems unnecessary in adult, non-pregnant horses with several previous EHV-1/4 exposures and should not be recommended. Overall, a more nuanced EHV-1/4 vaccination approach, taking the EHV-1 immune status of the horse into account, might be beneficial to achieve better protection of the horse population in countries facing EHV-1 and EHM outbreaks. While young horses or those with non-existing EHV-1 immunity likely need more frequent vaccination to reach stable circulating EHV-1 antibody levels, horses with robust existing immunity might be over-vaccinated with the current annual or 6-month vaccination intervals for EHV-1/4.

### 3.4. Serum Isotype Responses After EHV-1/4 Vaccination

In addition to total antibody responses, we evaluated the EHV-1 specific isotype response after frequent vaccination, which was dominated by IgG4/7 (Figure 3B). Anti-gC IgG4/7 was the only isotype present in serum prior to vaccination, increased immediately after V1, and then stayed high throughout the study (*p* < 0.0001), which largely mimicked the total antibody response in Figure 3A. In contrast, IgG1 only increased after V1, peaked at day 12 (*p* < 0.001), and then declined to low levels by 3 months. V4–V6 did not reactivate the IgG1 response and anti-gC IgG1 levels stayed low until the end of the study at 10 months. IgG3/5 antibodies stayed low after vaccination. Overall, these EHV-1 gC-specific isotype findings were similar overall to those found in the prior pregnant mare study [35]. IgG4/7 dominated the response, with a pattern mirroring total antibody levels, while IgG3/5 responses were low. However, IgG1 responses to vaccination with the inactivated vaccine were slightly different in pregnant mares that were EHV-1 naïve prior to their vaccination. Although the initial IgG1 peak after V1 was observed in both studies, the naïve mares also responded with IgG1 peaks to the vaccinations at 6 and 8 months [20], which were missing here in the non-pregnant, previously EHV-1 exposed horses. Equine IgG1 has been shown previously to increase rapidly after EHV-1 infection but is rather short-lasting during systemic EHV-1 antibody responses [15,16,28,29]. In contrast, and as shown here for d0/pre-V1 systemic EHV-1 antibody values, anti-EHV-1 IgG4/7 represented the long-lasting serum isotypes 20+ months after the last EHV-1 exposure. In previous studies, IgG4/7 provided long-term systemic immunity and protection against EHV-1 infection [28,29,36], while serum IgG1 was low or missing in horses that were fully protected against EHV-1 challenge infection [28,29]. Both EHV-1 specific IgG1 and IgG4/7 have recently been shown to effectively neutralize EHV-1 [36]. However, high IgG1 serum antibodies do not seem to be essential for protection in immune horses [28,29]. In summary, vaccination with the inactivated vaccine used here induced EHV-1 specific isotype profiles composed of short-lasting IgG1 and long-lasting IgG4/7. This serum isotype profile resembles the isotype response provoked after EHV-1 infection [15,16,28,29] and differences in the Prev-Vacc and Prev-Infect horse groups were also not observed here. It can thus be concluded that the inactivated vaccine re-activated and maintained an effective protective immune response against EHV-1 in horses with previous exposure to EHV-1.

### 3.5. Mucosal Antibody Induction After i.m. Vaccination

Next, we explored if i.m. vaccination stimulated an increase in antibodies at the viral entry site of EHV-1, the URT. Nasal mucosal epithelial cells are the first cells that are infected and where EHV-1 undergoes its initial viral replication cycles after transmission [8,36,42]. Thus, mucosal immunity at the URT provides an important first line of defense against EHV-1 [28,36,37,42]. Sufficient mucosal immunity can control the virus locally by significantly reducing or preventing disease, viral shedding, and cell-associated viremia [28,29]. In particular, mucosal IgG1 and IgG4/7 antibodies can neutralize EHV-1 and inhibit viral entry and replication at the URT [36].

In this study, horses still had mucAbs from their last EHV-1 exposure over 20 months ago before their first vaccination V1 (Figure 4A). Anti-gC mucAbs increased significantly after administration of V1 (*p* < 0.05), V4 (*p* < 0.01), and V6 (*p* < 0.001). They were also higher than pre-V1 antibody levels at 6 and 10 months (both *p* < 0.05). Similarly, anti-gB and gD mucAbs were elevated at several time points after V4 and V6 (Appendix A). At all timepoints after vaccination, mucAbs were clearly detectable, although overall they declined more rapidly than serum antibodies and tended to decrease after V3 and V5. In conclusion, frequent i.m. vaccination with the inactivated vaccine induced mucAb increases at the URT. MucAb induction after i.m. EHV-1/4 vaccination has not been investigated or described previously. Off-label intranasal vaccination, using an EHV-1 modified-live vaccine approved for i.m. injection, has sometimes been performed as an approach in EHV-1 outbreak situations to provoke mucosal immunity, but the efficacy of this practice is still unclear. Based on our current findings, i.m. vaccination appears to be a valuable method to increase mucosal EHV-1 specific antibodies locally at the URT. The increase in mucAbs at the URT after i.m. EHV-1/4 vaccination still needs to be confirmed for other EHV-1 or EHV-1/4 vaccine brands. In addition, it would be beneficial to evaluate the longevity of anti-EHV-1 mucAbs after less frequent vaccination than performed here, and to identify protective cut-off values for EHV-1-specific mucAbs, similar to those identified for serum. Overall, the quantification of mucAbs could become a valuable additional indicator of EHV-1/4 vaccine efficacy. 

### 3.6. EHV-1 Specific Mucosal Antibodies After i.m. Vaccination Are Dominated by IgG4/7 Isotypes

Like serum EHV-1 antibodies, anti-gC mucAb isotypes after EHV-1/4 vaccination were mainly composed of IgG4/7 (Figure 4B). Pre-existing anti-gC IgG4/7 antibodies were also present at the URT. Compared to pre-vaccination, mucosal IgG4/7 quickly peaked after V1 (*p* < 0.05), increased steadily by inspection after V2 and V3, and was also significantly increased after V4 at the 4m (*p* < 0.05) and 6-m sampling times (*p* < 0.001). After V5 at 6 m, mucosal anti-gC IgG4/7 decreased, to increase again after V6. However, between 7–10 months, mucosal anti-gC IgG4/7 was consistently higher than pre-vaccination levels at day 0 (*p* < 0.05). Nasal mucosal IgG4/7 were recently reported as the key antibody isotypes involved in neutralization of EHV-1, thereby preventing viral replication at the URT [36]. In addition, pre-existing mucosal IgG1 and IgG4/7 against EHV-1 were associated with protection from infection and disease [28]. Horses that were protected from infection showed a rapid increase in mucosal EHV-1 specific IgG4/7 during the first two days after experimental EHV-1 infection [28,29], interfering with the establishment of high and long-lasting nasal EHV-1 shedding after infection [28,29,36].

In our horses with pre-existing EHV-1 immunity prior to vaccination, mucosal anti-gC IgG1 antibodies were low prior to V1 (Figure 4B), slightly increased in response to V1 by day 12 (*p* < 0.05), then decreased and stayed low for the remainder of the study period, similar to the response observed in serum. Mucosal IgG3/5 isotypes were low (mean < 100 MFI) at all time points pre- and post-vaccination. Finally, mucosal anti-gC IgA stayed at a low, slightly varying level during the entire period of the study and was not influenced by i.m. vaccination (Figure 4B). It has recently been shown in horses that mucosal IgA had no or only a very low neutralizing capacity for EHV-1, while mucosal IgG1 and IgG4/7 were highly effective in neutralizing the virus. This was evaluated by purifying the individual mucosal antibody isotypes from nasal washes of horses after recovering from an experimental EHV-1 infection and then testing the individual mucosal isotypes in neutralization assays [36]. However, low baseline levels of mucosal IgA recognizing EHV-1 exist in most horses even before first EHV-1 exposure without majorly increasing after initial experimental infection [15] or challenge with EHV-1 [29]. This suggests that mucosal IgA against EHV-1 may have some role in the mucosal defense but there is no development of a robust effector or memory B-cell response against EHV-1 and IgA antibodies are likely of low affinity.

In summary, mucAb isotype profiles stimulated by i.m. vaccination with the inactivated EHV-1/4 vaccine used here mirrored the protective IgG4/7-dominated profile after EHV-1 infection. Therefore, mucosal IgG4/7 antibodies induced by vaccination are likely similarly effective in protection against EHV-1 infection and disease.

## 4. Conclusions

Our data presented in this frequent i.m. EHV-1/4 vaccination study resulted in two major findings. First, mucosal EHV-1 specific IgG4/7 antibodies were induced by the inactivated commercial vaccine used here. Mucosal immunity at the viral entry site plays an essential role in the protection against EHV-1 infection by minimizing viral replication at the URT. IgG4/7 antibodies are correlates of protection from disease, viral shedding, and cell-associated viremia. Therefore, EHV-1 or EHV-1/4 vaccines that induce mucosal IgG4/7 can effectively prevent infection and disease induced by EHV-1. Second, frequent EHV-1/4 vaccination of horses with pre-existing antibody immunity resulted in very high and stable serum antibodies against EHV-1, with isotype profiles mimicking EHV-1 infection immediately after the first vaccination booster. We found moderate to high pre-exiting antibodies resulting from EHV-1 or EHV-1/4 exposure more than 20 months prior to this current vaccination. The systemic antibody response suggested that these horses had robust EHV-1 immunity and B-cell memory. Overall, a more nuanced EHV-1 vaccination approach, taking the EHV-1 immune status of the horse into account, might be beneficial to achieve better protection of the horse population and minimize the number of EHV-1 and EHM outbreaks.

## Figures and Tables

**Figure 1 vaccines-13-00290-f001:**
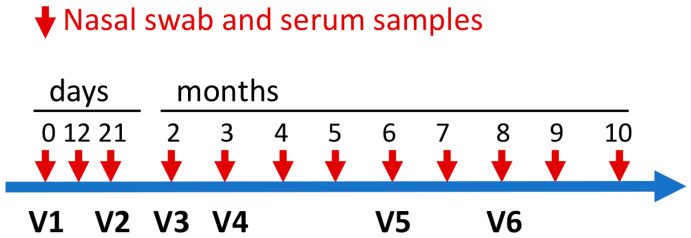
Study timeline of frequent EHV-1/4 vaccination and sampling. Horses with existing immunity from previous vaccination against EHV-1/4 (n = 6) or experimental infection with EHV-1 (n = 8) were repeatedly vaccinated with Calvenza EHV^®^ (V1–V6). The vaccine was given i.m. into the pectoral muscles. Serum and nasal swab samples for antibody measurement were taken immediately before each vaccination and at several additional time points for this study. The red arrows show all sampling times. A similar frequent vaccination approach with the same vaccine was previously performed in pregnant mares and resulted in decreasing serum antibodies after some of the vaccine administrations. In this study, we aimed to analyze if adult non-pregnant mares and/or geldings would respond similarly to repeated vaccination.

**Figure 2 vaccines-13-00290-f002:**
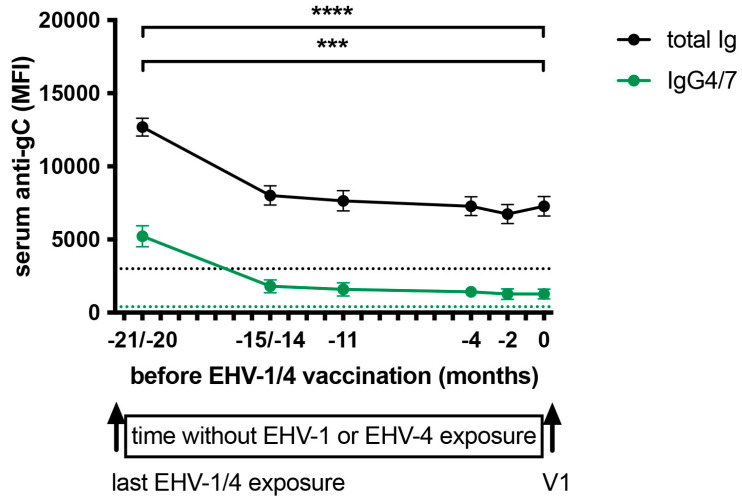
Anti-EHV-1 gC serum antibody decline in 14 adult horses in a 22-month period without EHV-1/4 exposure. The graph starts after the last EHV-1/4 exposure (arrow) and ends before the first vaccination (V1, arrow)) of the frequent EHV-1/4 vaccination approach described in this article. All horses had a complete EHV-1/4 vaccination and infection history. The last EHV-1/4 exposure of these horses was 22 or 21 months prior to the start of the frequent vaccination. Samples were taken at 21 or 20, 15 or 14, 11, 4, and 2 months, and at 0 month directly before V1. Serum antibodies against EHV-1 glycoprotein C (gC) were quantified in a fluorescent bead-based assay (EHV-1 Risk Evaluation assay). EHV-1 gC-specific total Ig and IgG4/7 antibodies were measured, and values are shown as median fluorescent intensity (MFI). The black dotted horizontal line shows the anti-gC total Ig cut-off value at 3000 MFI and the green dotted line at 400 MFI is the anti-gC IgG4/7 cut-off value. Antibody values above the cut-off value are indicative of a low infection risk if horses are exposed to EHV-1. *p*-values (*** *p* < 0.001, **** *p* < 0.0001) mark serum antibody decreases compared to antibodies at 21/20 months before V1. The color of the asterisks represents serum total Ig (black) or IgG4/7 (green). Brackets span identical *p*-value ranges.

**Figure 3 vaccines-13-00290-f003:**
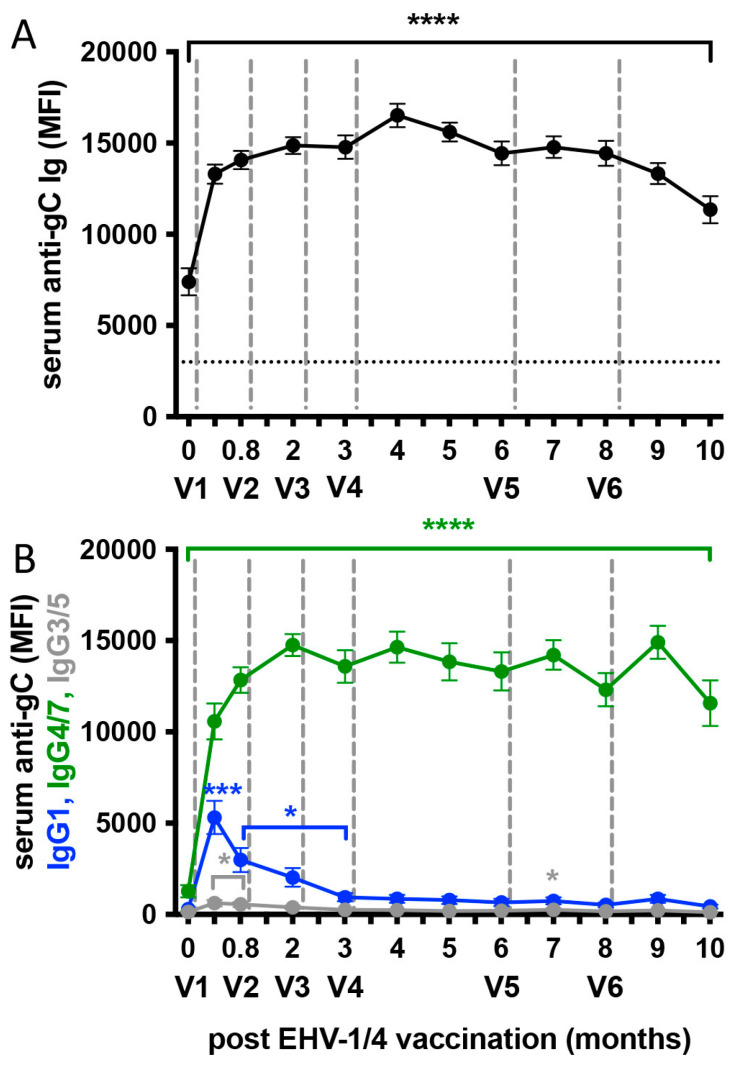
Serum antibody responses after frequent i.m. vaccination of horses with an inactivated EHV-1/4 vaccine. Adult horses (n = 14) with pre-existing immunity against EHV-1 were repeatedly vaccinated (V1–V6). Serum antibodies against EHV-1 glycoprotein C (gC) were quantified in a fluorescent bead-based assay (EHV-1 Risk Evaluation assay) at a serum dilution of 1:400. (**A**) EHV-1-specific total Ig and (**B**) IgG1, IgG4/7 and IgG3/5 isotypes in serum were measured as median fluorescent intensity (MFI). The gray dashed vertical lines show the times of vaccination after the serum sample were taken on the respective days. The dotted horizontal line in the upper graph shows the anti-gC total Ig cut-off value at 3000 MFI and antibody values above the cut-off value are indicative of a low infection risk if horses are exposed to EHV-1. *p*-values (* *p* < 0.05, *** *p* < 0.001, **** *p* < 0.0001) mark serum antibody increases compared to pre-V1 antibodies. The color of the asterisks represents the respective serum total Ig or isotypes. Brackets span identical *p*-value ranges.

**Figure 4 vaccines-13-00290-f004:**
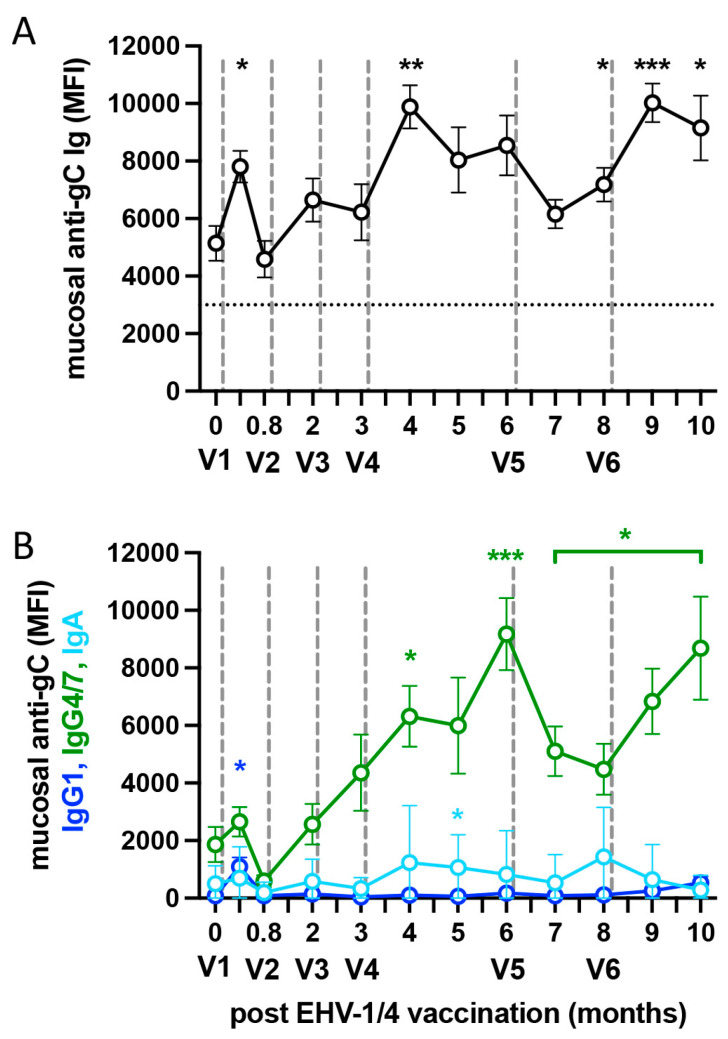
Nasal mucosal antibodies (mucAbs) after frequent EHV-1/4 vaccination. Horses with pre-existing immunity against EHV-1 (n = 14) were repeatedly vaccinated i.m. with an inactivated EHV-1/4 vaccine (Calvenza EHV^®^, V1–V6). MucAbs against EHV-1 gC were measured in undiluted nasal secretion samples using a fluorescent bead-based multiplex assay. (**A**) MucAb total Ig levels and (**B**) mucosal IgG1(blue), IgG4/7 (green), and IgA (sky blue) isotypes are shown as median fluorescent intensity (MFI). The dotted horizontal line in (**A**) represents the anti-gC total Ig cut-off value for serum. The dashed vertical lines mark the times of vaccine administration. *p*-values (* *p* < 0.05, ** *p* < 0.01, *** *p* < 0.001) show mucAb increases in comparison to mucAbs pre-V1 vaccination. The color of the asterisks represents the respective mucAb isotypes. The bracket spans IgG4/7 results with identical *p*-values.

**Table 1 vaccines-13-00290-t001:** Full EHV-1 infection and EHV-1/4 vaccination history of the horses (n = 14) prior to the frequent EHV vaccination approach described here.

Group	Horse ID	Age (Years)	Gender	EHV-1 History	Last Previous Vaccination or Infection
Previous EHV-1/4 Vaccination (Prev-Vacc)	M2	11	mares	annually vaccinated 2, 3, and 4 years prior to this approach ^a^	22 months ago
	M3	9			
	M4	10			
	M10	12			
	M11	12			
	M13	13			
Previous EHV-1 Infection (Prev-Infect)	F3-1	5	geldings	experimental EHV-1 infection at 2.5 and 3 years of age ^b,c,d^	21 months ago
	F4-1	5			
	F7-1	5	mare		
	F11-1	5	geldings		
	F12-1	5			
	F2-1	5	mares	experimental EHV-1 infection at 3 years of age ^b,d^	
	F6-1	5			
	F10-1	5	gelding		

^a^ Horses were imported from Iceland 4 years prior to this study. In the US, they were then vaccinated against Equine Rhinopneumonitis (Calvenza EHV), as previously described [35]. ^b^ Horses were imported from Iceland 3 years prior to this study. They were 22 months of age at the time of importation to the US. ^c^ Horses were experimentally infected at 2.5 years of age with EHV-1 [16]. ^d^ Horses were experimentally infected with EHV-1 Ab4 at 3 years of age [28].

## Data Availability

All data generated and analyzed during this study are included in this published article and its Appendix A.

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
