# Peer review of "Increase in Virus-Specific Mucosal Antibodies in the Upper Respiratory Tract Following Intramuscular Vaccination of Previously Exposed Horses Against Equine Herpesvirus Type-1/4"

_vaccines, 2025, doi:10.3390/vaccines13030290_

Round 1
Reviewer 1 Report
Comments and Suggestions for Authors
This is an excellent practical study utilising a currently available vaccine with confirmation of the importance of the IgG4/7 mucAb response
The manuscript would benefit by attending to a few areas for greater clarity to the reader rather than relying on numerous assumptions:
Including amending “EHV s” throughout manuscript including in title this is sloppy as it includes gammas -best to ensure specific ID of eg EHV-1, EHV-1/4 where appropriate
Perhaps a little more detail of Calvenza vaccine virus in Materials would help. Particularly worth discussing in results /discussion that this is an I/M vacc able to boost mucosal immunity not often reported in respiratory viruses
Also greater detail/ discussion of aspects of this study particularly with respect to divergence from conventional accepted practices :
see attached file for further details

Reviewer 2 Report
Comments and Suggestions for Authors
This is one of a series of studies on the prevention of EHV-1 infection by Wagner et al. This paper set two goals: (1) whether the i.m. vaccine application route induces mucosal antibodies against EHV-1, and (2) if non-pregnant, adult horses (mares and geldings) also produce short-lasting systemic antibody responses after frequent vaccination. Both objectives were achieved to some extent. The content of the paper is excellent and will make a significant contribution not only to EHV-1 research but also to equine health and safety. The description of the paper is scientific and precise. Several questions and comments are raised, as follows.
L193: Reference 23 does not contain the method in detail. Reference 20 contains the detail of the method used here. Clarify references cited.
L226-231: Is EHV-4 infection not a problem?
L233-243: How high were the antibody titers after the vaccination before this experiment? Has it declined since then, or has the same level of antibody titer been maintained, which was measured in this experiment at the time of V1? It would be useful to present the data here.
It should also be described that after V4, the antibody titer decreased with a peak at 4 M. There appears to be no effect of V5 and V6.
L255: There is no point in showing P-values in 3 steps. Why is it necessary to show three levels? It does not make sense from a statistical point of view.
L283-314: If we consider most Ig to be IgG4/7 after 3M, why is IgG4/7 not consistent with serum Ig titer and variation? Is there any consideration as to how much other subclasses are considered to be affected?
Also, the titers decreased after v3, but increased after a month for V4, V5, and V6, and then decreased after that. Thus, the effect of additional vaccinations appears to be unstable. Furthermore, the booster effect appears to be weak or almost nonexistent. Can we really say that the inactivated vaccine re-activated and maintained an effective protective immune response against EHV-1 in horses with previous exposure to EHV-1? -1?
L344-345: total Ig is IgG only.
Here, IgA data should be included in Figure 3 and not Figure 4.
L369-370: If we consider most Ig to be IgG4/7 after 3M, why do the serum Ig titers and variations not match the IgG4/7? Is there any consideration as to how much other subclasses are considered to be affected?
Is the amount of IgG1 and IgG4/7 equal to that of serum Ig?
Should the IgA data be included in Figure 3.
Or perhaps it would be better to show Figures 3 and 4 together.
Reviewer 3 Report
Comments and Suggestions for Authors
In this study, by repeatedly immunizing both experimentally infected horses and immune horses with inactivated EHV vaccine, the anti-EHV antibody of immune horses was maintained above the critical value of EHV infection, providing good immune protection for immune horses.
However, the following problems were also found:
- To illustrate the difference between immunized and pre-immunized horses and immunized and previously immunized horses, an experimental group should be added: uninfected and unimmunized horses;
- In order to explain the difference between antibodies immunizing pregnant mares and non-pregnant mares, another experimental group should be added: pregnant mares;
- According to the antibody test results, 21-22 months after immunization or infection, the antibody is still far above the critical value, and the antibody before each immunization is also far above the critical value. The paper also proposed that the immunization interval should be set at 6 months or one year, which may actually reach more than 18 months (21-22 months should be no problem), please explain why the immunization interval is designed so short. So many immunizations? As many as six?
- Please explain why the antibodies of mucosal species did not increase after the fifth immunization, but decreased for 2 consecutive months. At the 6th month of the test, the single antibody basically increased, while the total antibody gC decreased or remained flat. Is the antibody determination method inaccurate?
- When pregnant mares are vaccinated according to the same vaccination procedure, the antibody drop after 2 months and 8 months of vaccination, the antibody drop after 1-2 weeks of vaccination is understandable, if the antibody drop is still longer after vaccination, it is not normal, please explain why.
Round 2
Reviewer 2 Report
Comments and Suggestions for Authors
I appreciate your answers and explanations. The manuscript is crucial to our understanding and controlling the equine herpesvirus 1 infection.